# Stabilization of Acne Vulgaris-Associated Microbial Dysbiosis with 2% Supramolecular Salicylic Acid

**DOI:** 10.3390/ph16010087

**Published:** 2023-01-08

**Authors:** Hazrat Bilal, Yuanyuan Xiao, Muhammad Nadeem Khan, Jinyu Chen, Qian Wang, Yuebin Zeng, Xinyu Lin

**Affiliations:** 1Department of Dermatology, The Second Affiliated Hospital of Shantou University Medical College, Shantou 515041, China; 2Department of Dermatology, The Affiliated Hospital of Southwest Medical University, Luzhou 646000, China; 3Department of Dermatology, Sichuan Provincial People’s Hospital, University of Electronic Science and Technology of China, Chengdu 610072, China; 4Chinese Academy of Sciences, Sichuan Translational Medicine Research Hospital, Chengdu 610072, China; 5Department of Dermatology, People’s Hospital of Deyang City, Deyang 618000, China; 6Faculty of Biological Sciences, Department of Microbiology, Quaid-I-Azam University, Islamabad 45320, Pakistan; 7Department of Dermatology, Chengdu Second People’s Hospital, Chengdu 610017, China; 8Department of Medical, Surgical and Experimental Sciences, University of Sassari—Neurology Unit, Azienza Ospedaliera Universitaria (AOU), 07100 Sassari, Italy

**Keywords:** acne vulgaris, supramolecular salicylic acid, high-throughput sequencing, facial microbiota

## Abstract

Facial microbiota dysbiosis is an important factor in causing acne vulgaris. The present study aimed to analyze the effect of 2% Supramolecular Salicylic Acid (SSA) on acne-associated facial bacteria. In the current study, 30 acne vulgaris patients (treated with 2% SSA for eight weeks) and ten volunteers with no facial acne were selected. Samples from acne patients (before and after treatment) and volunteers (not treated) were analyzed via high throughput sequencing, Deblur algorithm, and R microbiome package. After treatment with 2% SSA, the total lesion count and global acne grading system (GAGS) score reduced significantly (*p* < 0.001). Metagenomic sequencing analysis revealed that the pre-treated acne group had low α and deviated β diversity compared to the control and post-treated acne groups. Due to the treatment with 2% SSA, α diversity index was increased and β diversity was stabilized significantly (*p* < 0.001). The relative abundance of bacterial genera in the pre-treated acne group was uneven and had a high proportion of *Staphylococcus*, *Ralstonia,* and *Streptococcus*. The proportion of these three genera was significantly decreased in the post-treated group, and overall bacteria genera distribution tends toward the healthy individual. It is concluded that 2% SSA normalizes the microbial communities associated with the skin.

## 1. Introduction

Acne vulgaris (acne) is a chronic inflammatory dermatosis of the sebaceous glands. The morbidity rate is highly reported among the youth, i.e., 70–87%, but in recent years its prevalence in adults has also risen [1,2]. The incidence of acne is associated with different skin types, such as being highly prevalent in oily (61.23%), followed by neutral (36.28%) and dry skin (36.09%) [3]. The signs of acne depend on the severity of the conditions, most commonly characterized by closed and open comedones, inflammatory papules, nodules, pustules, and cysts [4]. Severe acne form scars, which can change the skin’s appearance negatively, especially in visible areas such as the face. These features of the disease cause anxiety, depression, and self-abasement, which affect a person’s mental state and social activity [5].

The pathogenesis of acne depends on various factors, including sebum over-secretion, follicular hyper-keratinization, inflammatory responses, and the role of bacteria [6]. These factors are interconnected, such as due to the increased sebum production where many lipophilic bacteria like *Cutibacterium acnes* proliferate. These bacteria produce free fatty acids due to the lipase enzyme, which provokes the cells to secrete IL-1α and other cytokines to induce inflammatory responses and contribute to acne production [7]. According to earlier studies, *C. acnes* were responsible for acne; however, recent studies have shown that dysbiosis of facial microbiota plays a significant role in acne production [8,9].

Some recent studies stated the promising anti-acne therapeutic efficacy of oral probiotics, oxybrasion, and synergistic treatment protocol of hydrogen purification and cosmetic acids [10,11,12]. However, these studies are preliminary and further validations are required to prove their safety and efficacy. Currently, dermatologists use various treatment approaches, i.e., retinoids, antibiotics, anti-androgen, and salicylic acid to cure acne [13]. Salicylic acid is one of the peeling agents and is effective in treating acne due to its keratinolytic, anti-inflammatory, and bactericidal effects [14]. However, its application is limited due to its poor solubility in water and forming crystals in low-pH alcoholic solutions, which cause skin irritation [15]. To overcome this limitation, a supramolecular technique based on a reversible and non-covalent bonding approach was applied to develop an intramolecular complex. This results in water-soluble supramolecular salicylic acid (SSA) which has the properties of slow release upon application, efficiency in low-pH, and low skin irritation [2,5].

As mentioned earlier, microbial imbalance on the skin surface leads to acne production, and SSA is currently used for their treatment; however, their effect on acne-associated facial microbiota is not fully explored [2,8]. Therefore, in the current study, we aimed to evaluate the impact of 2% SSA on the composition and diversity of facial skin microbiota of acne patients using the high-throughput sequencing approach. The findings will provide information about the effect of 2% SSA on the facial-bacterial communities and new ideas for acne vulgaris treatment.

## 2. Results

### 2.1. Total Lesion Count and GAGS Score Comparison before and after Treatment

The total lesion count and GAGS score after treatment in the acne group were significantly lower than before treatment with *p* < 0.0001 (Table 1). Pictures of the patients before and after treatment are also provided in Appendix A.

### 2.2. Observation of Adverse Reactions

Two patients had dry skin desquamation on the third and fourth days of the treatment; the effect was mild, and the patients were advised to increase moisturizing time and pay attention to sun protection. The adverse effects recovered on their own after one week of self-care.

### 2.3. Metagenomic Results

The sequencing depth was uniform throughout the samples and sufficient for analysis and comparison (Appendix A. From the sequencing of 70 samples, 2,407,012 raw reads were obtained that were purified in 2,361,048 reads of 250-bp length after sequence joining, filtration, and noise removal. These sequence reads were clustered into 2557 ASVs, and the representative sequences of ASV were annotated by the Naive Bayes method, which was annotated into 33 bacterial phyla, 2345 classes, 2320 orders, 2256 families, and 1805 genera. Moreover, the pre-treatment acne group carried a mean of 994 ASVs, the treated group carried a mean number of 1574 ASVs, while the control group carried a mean number of 982 ASVs.

### 2.4. Effect of 2% SSA Treatment on Microbial Diversity

It was observed that 2% SSA treatment had changed bacterial α and β microbial diversity among acne patients.

#### 2.4.1. Alpha Diversity

The α diversity based on the Chao1, Simpson, Shannon, and phylogenetic diversity (Faith’s PD) index increased in response to the 2% SSA treatment compared to pre-treated and control groups. The Chao1 and Faith’s PD before acne treatment were lesser than those in the treated and control groups, but the difference was not statistically significant. On the other hand, α diversity measured on Shannon and Simpson indexes is higher in the treated and control group when compared to the pre-treated group. This indicates that microbial richness and uniformity are significantly lower in the pre-treated acne group compared to the treated and control groups. The detailed results are presented in Figure 1 and Appendix A.

#### 2.4.2. Beta Diversity

It was observed that the β diversity of the bacterial population changed in response to the 2% SSA treatment. The β diversity of the pre-treatment acne group has a deviated pattern compared to the treated and control groups, while β diversity in the treated group is very close to that of the control group. The statistical significance among the three groups was determined with Analysis of Similarity (ANOSIM) and Permutational Multivariate Analysis of Variance (PERMANOVA), adjusted for pre-treated, treated, and control groups. The *p*-value equal to or less than 0.05 was considered statistically significant, and it was observed that the change between the pre-treated acne group and the treated acne group was significant (*p* < 0.001), but non-significant between the treated acne group and the control group (Table 2). The results showed that the microbial communities on the skin surface of the acne patients were significantly different from that of the healthy individuals. The results also indicate that the microbial communities of the acne patients have changed significantly in response to the 2% SSA treatment (Figure 2).

### 2.5. Effect of 2% SSA on Bacterial Communities

It was observed that 2% SSA had changed the microbial communities of acne patients. The changes were not significant at the phylum level, but the bacterial communities of the pre-treated and treated groups were quite different at the genus level.

### 2.6. Diversity at the Phylum Level

At the omission point of 0.01%, the pre-treated acne group carried 16, the treated group carried 23, and the control group carried 20 phyla. The most abundant phyla in the three groups were *Proteobacteria, Firmicutes, Actinobacteria,* and *Bacteroidetes*. The phylum *Proteobacteria* accounted for 53% of the pre-treatment acne group, 52% of the post-treatment acne group, and 53% of the control group. The *Firmicutes* accounted for 36% of the pre-treatment acne group, 26% of the post-treatment group, and 24% of the control group. *Actinobacteria* accounted for 8% of the pre-treatment acne group, 13% of the post-treatment acne group, and 15% of the control group. The phylum *Bacteroides* accounted for 1.96% of the pre-treatment acne group, 7% of the post-treatment acne group, and 6% of the control group. The average relative abundance of the remaining phyla in the three groups did not exceed 1%. The difference among the groups at the phylum level was non-significant. Detailed results are presented in Figure 3 and Appendix A.

### 2.7. Diversity at the Genus Level

The 2% SSA treatment affected the composition of microbial communities at the genus level. When an omission point was adjusted to 0.01%, it was observed that the pre-treated acne group contained a mean number of 208 bacterial genera, the treated group contained 323 genera, and the control group carried 298 bacterial genera. The dominant genera in the pre-treatment acne group were *Ralstonia* (37%), and *Staphylococcus* (29%), while the relative abundance of the remaining genera did not exceed 5%. The genera with the highest relative abundance after treatment in the acne group were *Staphylococcus* (17%), *Ralstonia* (13%), *Phreatobacter* (10%), and *Lawsonella* (7%), while relative abundance of the remaining genera did not exceed 5%. The top genera with the highest average relative abundance in the control group were *Phreatobacter* (14%), *Staphylococcus* (13%), *Ralstonia* (5%), and *Streptococcus* (4%). Relative abundance of the remaining genera was less than 5%. Three genera such as *Phreatobacter, Ralstonia,* and *Staphylococcus,* had a significant difference (*p ≤* 0.05) among the three groups. The relative abundance of *Phreatobacter* was significantly low in the pre-treated acne group compared to the treated and control groups. The relative abundance of *Ralstonia* and *Staphylococcus* were significantly higher in the pre-treated group when compared to the treated and control group. The detailed results are presented in Figure 4 and Appendix A.

### 2.8. The Pattern of Microbiota Changes in Response to 2% SSA Treatment

When the results were visualized taxonomically it was observed that the three groups are mainly inter-connected in the phylum *Proteobacteria*, followed by *Firmicutes* and *Actinobacteria*. From the sector area occupied by each group it can be observed that the relative abundance of the *Proteobacteria* phylum of the three groups is similar. While *Firmicutes* in the pre-treated acne group were significantly higher than in the other two groups, and *Actinobacteria* were significantly lower than in the other two groups. The proportion of bacteria belonging to class *Alphaproteobacteria,* order *Rhizobia, and* family *Rhizobiales incertae sedis* was significantly lower in the pre-treated group than in the control and treated groups. The taxonomy chart showing the proportion of bacteria from phylum to genus reported in the three groups of samples is presented in Figure 5. Furthermore, from analysis of phylum *Firmicutes,* it was observed that the genus *Streptococcus* remained almost in equal proportion in all three groups. In contrast, the proportion of *Staphylococcus* was much higher in the pre-treated group, but after treatment its burden tends to be like the control group.

When using the snakey chart it was observed that the relative abundance of *Ralstonia* and *Staphylococcus* were high in the pre-treated group, while their ratio significantly decreased after treatment. In contrast, the *Phreatobacter* in the treated group was significantly higher than in the pre-treated group. The relative abundance of bacterial genera in control and 2% SSA-treated groups is comparatively look-alike than the pre-treated group (Figure 6A).

From cluster analysis it can be observed that the microbial communities of each sample in the control group were close to some samples of the treated group. Similarly, some of the samples from the pre-treated group were close to the treated group; however, significantly less proximity was reported between the samples of the pre-treated and control group (Figure 6B).

Further, to see which bacterial genera are significantly adapted in the acne group and which genera are significantly changed in response to 2% SSA treatment, LEfSe (Linear discriminant analysis Effect Size) was used. LEfSe explains differences between groups by coupling standard tests for statistical significance with additional tests encoding biological consistency and effect relevance. There were significant differences in bacterial composition relative abundance among the three groups at the phylum and genus levels. LEfSe results (Figure 7, LDA > 3.5) presented that the treated and control group were enriched by *Ralstoni, Staphylococcus,* and *Phreabacter.*

### 2.9. Core and Unique Microbiota

The pre-treatment acne carried a total of 994 ASVs; the treated group carried 1574 ASVs, and the control group carried 982 ASVs. Among the total collective ASVs, 356 were shared among three groups, 586 ASVs were shared between the pre-treated and treated group, 417 ASVs were shared between the pre-treated and control group, and 550 ASVs were shared between the treated acne group and the control group. Among the total ASVs, 347 were unique to the pre-treatment acne group, 794 ASVs to the treated acne group, and 371 to the control group. The number of ASV in the treated acne group was significantly higher than that in the control and pre-treatment acne groups. Similarly, the number of treated groups ASVs shared with the control group was higher than all other groups shared ASVs (Figure 8). It indicates that the pre-treated group microbial diversity is quite different, and upon treatment with 2% SSA it shifts closer to the control group.

## 3. Discussion

Measurement of acne severity in terms of diagnosis, follow-up, and effect of the specific medication is essential in clinical practice [16]. One commonly used method is lesion counting, representing a more accurate approach. Among lesion-counting methods, the global acne grading system (GAGS) is one of the most detailed systems that maintain simplicity [17]. In the current study, based on the GAGS score, 30 moderate acne vulgaris patients were treated with 2% SSA. The results revealed the total number of facial lesions and the GAGS score of treated patients was reduced significantly in response to 2% SSA treatment. Only two patients had mild dryness and desquamation adverse reactions and recovered independently. The efficacy of 2% SSA in the current study was in line with previously reported studies; however, more clinical trials against mild and severe acne are needed to confirm its efficacy and safety [2]. Studies showed that acne vulgaris mainly occur due to dysbiosis in the facial microbiota. Therefore, we analyzed facial microbiota of acne patients in response to 2% SSA treatment. The conventional culture technique for microbial diversity identification is unreliable due to the difference in growth conditions of some bacteria in natural and laboratory environments. Therefore, the 16S rDNA high throughput sequencing technique was used to determine the diversity and type of bacteria on the subject and control faces, which is more accurate and can find even the unculturable bacteria. Moreover, this study uses ASV instead of OTU for marker-gene analysis to minimize the amplification and sequencing error and to distinguish sequence variants up to only one nucleotide difference [18].

Human healthy skin produces proteases, isoenzymes, and antimicrobial peptides, inhibiting pathogenic bacteria colonization. However, the specific pH, temperature, humidity, and secretion of sebaceous glands allow the growth of skin normal-flora. While acne is associated with an imbalance (dysbiosis) of the facial microbiota [19] The present study observed that 2% SSA beneficially modulated the facial microbiota by promoting diversity and evenness in the bacterial communities. The 2% SSA increased the number of ASVs, bacterial phyla, and genera in the acne group. From the analysis of α and β microbial diversity, it was concluded that 2% SSA efficiently normalizes the diversity of skin microbiota. Different studies on microbiota characterization stated that a comparatively diverse bacterial group is considered a healthy microbiota [20].

Based on phylum diversity, the relative abundance in the post-treated and control group were relatively close to each other compared to the pre-treated acne group; however, they were not statistically significant. The abundance of different phylum in the current study was consistent with previous research, indicating that these are the residential phyla of the face [21]. Furthermore, the abundance of *Proteobacteria* and *Firmicutes* were high, and *Actinobacteria* was lower in the pre-treated group than in the post-treated and control group. Other studies report a similar trend of phyla in association with acne, showing that bacteria belonging to *Proteobacteria* and *Firmicutes* are mainly involved in acne infection [22,23]. At genus level it was found that *Ralstonia, Staphylococcus,* and *Phreatobacter* were found in high numbers in the three groups. However, their abundance was much higher and statistically significant in the pre-treated acne group compared to the treated and control groups. The cumulative abundance of *Ralstonia and Staphylococcus* in the pre-treated acne group was dominant (>65%), causing an uneven distribution of other normal flora, resulting in a micro-ecological imbalance which is the main reason for acne vulgaris [24]. It is worth noting that this study found that the relative abundance of *Cutibacterium* was only 0.1% in the pre-treated acne group and 0.5% in the control group. A recent study reported a similar abundance of *Cutibacterium* in healthy and acne patients [25]. Another study reported slightly higher *Cutibacterium* levels in healthy subjects, similar to our results [23]. The *Cutibacterium* was previously thought to be the main reason for acne development; however, recent studies identify the number of pathogenic bacteria like *Staphylococcus Faecalibacterioma, Streptococcus*, *Enterobacter*, *Klebsiella, Odirobacter,* and *Bacteroides* genera in association with acne vulgaris [26]. In the current study the type of bacterial genera in the acne and control group were comparatively similar, but their relative abundance in the acne group was uneven, while the control group was relatively balanced. A recent study stated that acne pathogenesis depends on the balance of bacterial species and their metagenomic elements [21]. The genus *Ralstonia* in the current study was found in high proportion in the pre-treated acne group, which is previously reported in association with atopic dermatitis [27]. However, their role in acne is still not identified and needs to be studied in future research.

The Sankey plot analysis of the three groups further clarifies the microbial shift of the pre-treated group toward the control group after treatment with 2% SSA [28]. It was observed that three bacterial genera such as *Staphylococcus, Ralstonia,* and *Phreatobacter,* significantly changed in response to 2% SSA treatment. Based on specific taxonomic classification from phylum to genus, the Firmicutes and Actinomycetes members significantly modulate in response to 2% SSA treatment. A specific taxonomic tree was used to observe changes from phylum to species level [29]. When the changes in facial microbiota were analyzed through LEfSe, it also confirmed the changes evaluated through the Sankey plot and specific taxonomic tree. LEfSe is mainly used to explain differences between classes by combining standard tests for statistical significance with additional tests encoding biological consistency and effect relevancy [30].

When the facial microbiota of the three experimental groups was assigned to the core and unique microbiota, it was observed that a high percentage (91%) of microbiota belonged to the microbiome. Studies have stated that a relatively high percentage of microbiota belonging to specific ecological sites in the same group acts as core microbiota [31]. Studies also suggest that disease is often caused due to unique microbiota, and in most microbiota modulation trials, the core microbiota remains unchanged [32,33]. Same was with 2% SSA treatment that changed the unique microbiota in the pre-treatment group, which was mainly responsible for dysbiosis.

The present study is the first of its kind that examines the effect of 2% SSA on acne-associated facial microbiota. However, some limitations in the present study might be responsible for biased outcomes. Such as, only grade II and III, indicating moderate acne, were selected for evaluation of the 2% SSA effect, while grades I and IV indicating mild and severe acne, respectively, were not selected. Moderate acne was selected because it is more consistent and usually present in some form or another [34]. Similarly, only 30 acne patients were selected from only one hospital. Therefore, further studies are required to validate the effect of 2% SSA on facial microbiota in different acne types and geographical locations.

## 4. Materials and Methods

### 4.1. Subject and Selection Criteria

According to the Pillsbury grading system, 30 patients having moderate acne (grade II and III) were selected from the department of dermatology at Sichuan Provincial People’s Hospital, China [35]. The chief dermatologist (X.L) performed patients’ diagnoses and grading. The mean age of the patients was 22 years (range; 18–30), including 12 male and 18 female cases. All the patients were treated with 2% SSA gel (twice a day, morning and evening) and a nourishing conditioning mask (once at night for 15 min) purchased from Shanghai Ruizhi Pharmaceutical Technology Co., Ltd. All subjects were advised to wash their faces 30 min before applying 2% SSA with water. None of the acne or control subjects could clean their face 24 h before sampling. The acne group was named (BT) before treatment and (AT) after the treatment. Ten volunteers—individuals with no facial acne—were selected as the control group (C), including four males and six females, and the age range was between 18 and 30 years. Two months before sampling, the subjects and control had not taken any oral or topical antibiotics, glucocorticoids, probiotics, yeast or sulfur tablets, or any other cosmetologically treatment. All of the subjects were advised not to take any make-up during treatment. The exclusion criteria included that none of the subjects and control had a pregnancy, were planning to have children, lactation, cosmetic or drug-induced acne, pro-inflammatory pigmentation, and scarring.

### 4.2. Data and Samples Collection

The number of lesions and data about the global acne grading system (GAGS) score were obtained from the patients before and after one day of 8 weeks of treatment [36]. The samples for molecular experiments were collected through swabs using the wetting solution (0.1% Tween-20 and 0.9% NaCl). The swabs were wiped up and down twenty times in the position of the face (6 × 6 cm^2^) that had multiple numbers of acne. The area from which the initial sampling was taken was noted, and sampling was performed from the same position after treatment. For the control group, the cheeks region was selected for sample collection. All the procedures were performed in sterile conditions, and samples were immediately stored at −80 °C until further processing.

### 4.3. The Amplified DNA Extraction, Amplification, and High-Throughput Sequencing

At the end of sampling, a total of 70 samples were collected, i.e., BT; *n* = 30 (pre-treatment acne group), AT; *n* = 30 (post-treatment acne group), and C; *n* = 10 (control). Genomic DNA was extracted and purified from all samples according to the kit manufacturer’s protocol (Zymo Research Co., Ltd. Irvine, CA, USA), following a preliminary step of bead-beating for 2 min. The integrity and concentration of DNA were detected following the Picogreen assay protocol [37]. The V3-V4 region of 16S rDNA was amplified using the primers pair; 515F:5′-GTGYCAGCMGCCGCGGTAA-3′ and 806R:5′-GGACTACHVGGGTWTCTAAT-3′ and reaction conditions described earlier [38]. The amplified product was recovered using Zymoclean™ Gel DNA Recovery Kit; quantified by Qubit 2.0 Fluorometer (Thermo Scientific, Waltham, MA, USA) and pooled in equimolar quantities. Subsequently, DNA library was constructed using the NEBNext^®^ Ultra™ II DNA Library Prep Kit for Illumina^®^ (New England Biolabs (UK) Co., Ltd. Hitchin, UK). High-throughput sequencing was performed using the llumina’s HiSeq PE250 sequencing platform. The kit used for sequencing was HiSeq Rapid SBS Kit v2 (500 cycles).

### 4.4. Data Analysis

FLASH software was used to merge the double-ended sequences, followed by the sabre tool to separate raw reads based on barcode. QIIME2 bioinformatics pipeline filters out sequences with an average quality below 30, length < 200-bp, and with a fuzzy base (N) number greater than zero. Denoised and chimera were removed from the sequence based on the Deblur algorithm to generate the ASV feature tables and feature sequences. A Naïve Bayes algorithm was used to construct a species classification dataset for the SILVA database, and the dataset was used to annotate species on the ASV feature sequences. Multiple alignments of feature sequences were performed, and phylogenetic trees were constructed using the FastTree tool. Each sample was homogenized and resampled based on the least amount of data in the sample. Alpha and Beta diversity were performed using the R language, and data visualization was performed using the ggplot2 package.

### 4.5. Reference Standard

In addition to the samples ZymoBIOMICSTM Microbial Community DNA Standard (D6305) https://www.zymoresearch.com/collections/zymobiomics-microbial-community-standards accessed on 20 September 2022) was added as an internal reference. This mock community comprises the genomic DNA of eight bacterial species. For the experimental and analytical procedures, the same methods were used as for the samples; the bacterial compositions are shown in Appendix A. Similarly, the reaction mixtures having ddH_2_O instead of samples were used as a negative control.

### 4.6. Statistical Analysis

The Student *t*-test and chi-square test were used to determine the significant difference between the parametric data. For the non-parametric data, the Kruskal–Wallis test, Wilcoxon Rank Sum Test, and Duncan’s test were used to determine the statistical significance. The *p*-value ≤ 0.05 were considered statistically significant. The linear discriminant analysis (LDA) effect size (LEfSe) algorithm was used to recognize significantly different species between groups (LDA scores > 3.5 k). The software and tools used in the current study were SPSS (*v*.25), QIIME2 (2020.2), FLASH (1.2.11), R language (4.0.5), Python (3.7.4), and SILVA database (138).

## 5. Conclusions

The current study analyzed the facial bacteria of 30 moderate acne vulgaris patients and their response after the treatment with 2% SSA. It was found that dysbiosis in facial microbiota can lead to acne development. The 2% SSA treatment significantly improved the GAGS score of moderate acne patients and very few adverse effects were noted in only two patients, which were recovered by themselves. The metagenomic analysis of the pre-treated and treated group concluded that the 2% SSA modulates the microbial diversity of patients toward like individuals with no facial acne. However, due to the relatively small sample size and selection of only moderate acne patients, this study suggests that more studies are necessary to draw a definitive conclusion on this topic.

## Figures and Tables

**Figure 1 pharmaceuticals-16-00087-f001:**
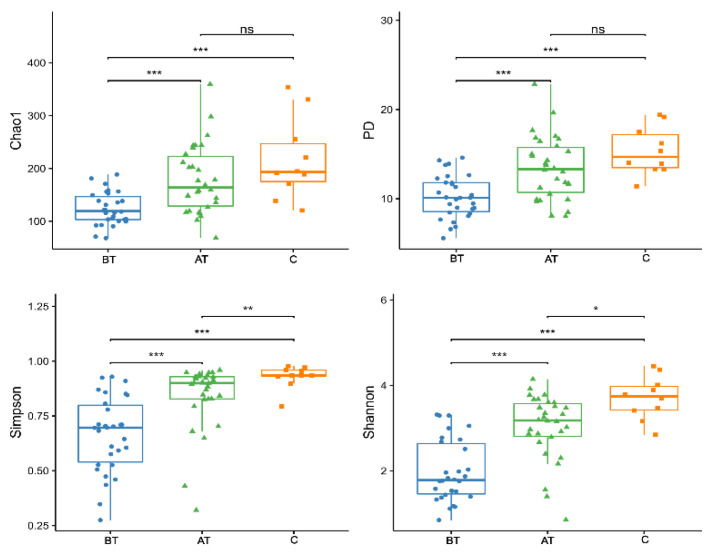
Alpha diversity indexes (Chao1, Simpson, PD, and Shannon) of the pre-treated (BT), after-treatment (AT), and control (C) groups. ns = no statistical significance, * = *p* <0.05, ** = *p* <0.01, *** = *p* < 0.001.

**Figure 2 pharmaceuticals-16-00087-f002:**
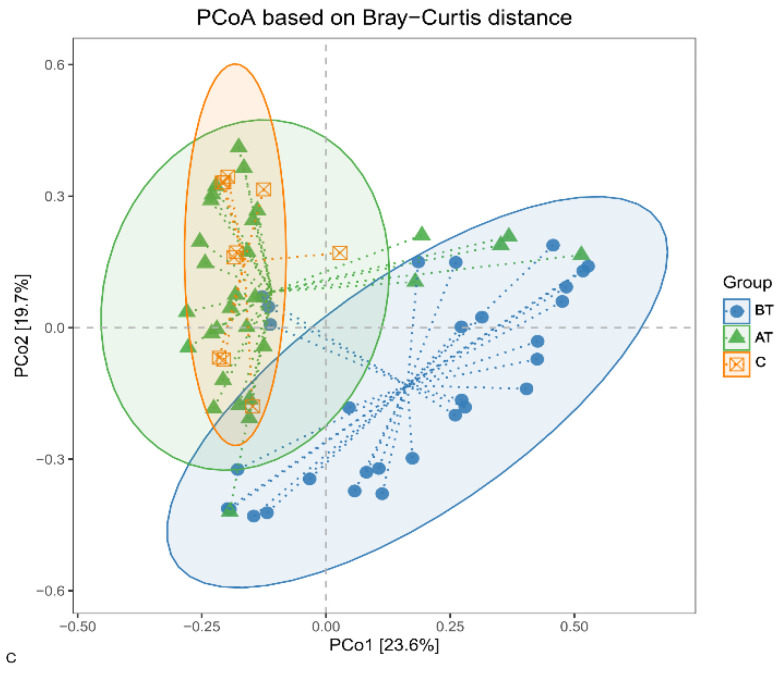
β diversity of facial microbiota of the pre-treated (BF), treated (AT), and control (C) acne groups. Principal Coordinates Analysis (PCoA) based on the Bray–Curtis distance matrix showed a marked separation between the microbial communities of BT and AT group, between BT and C group, in the case of AT and C group degree of separation is comparatively less.

**Figure 3 pharmaceuticals-16-00087-f003:**
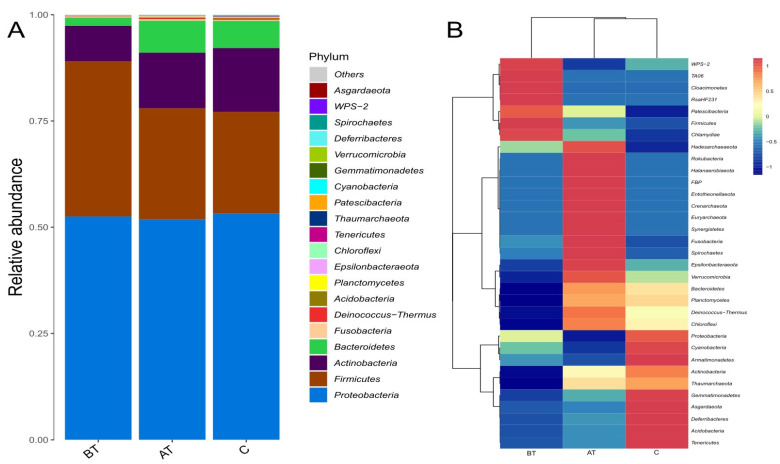
Relative abundance of facial microbiota at phylum level among the pre-treated (BT), treated (AT), and (C) acne groups (**A**): Barplot presenting mean relative abundance in terms of percentage among the three experimental groups. (**B**): The correlation heatmap presents relative abundance in percentage among the three experimental groups at the phylum level.

**Figure 4 pharmaceuticals-16-00087-f004:**
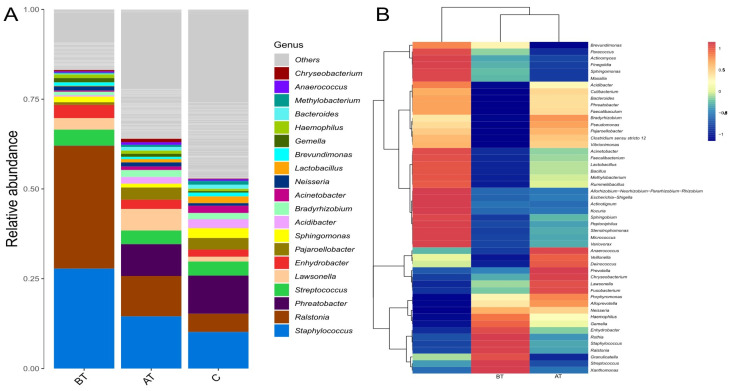
Relative abundance of facial microbiota at genus level among the pre-treated (BT), treated (AT), and control (C) groups. (**A**): Barplot presents relative abundance in percentage among the three experimental groups. (**B**): The correlation heatmap presents relative abundance in percentage among the three experimental groups at the genera level.

**Figure 5 pharmaceuticals-16-00087-f005:**
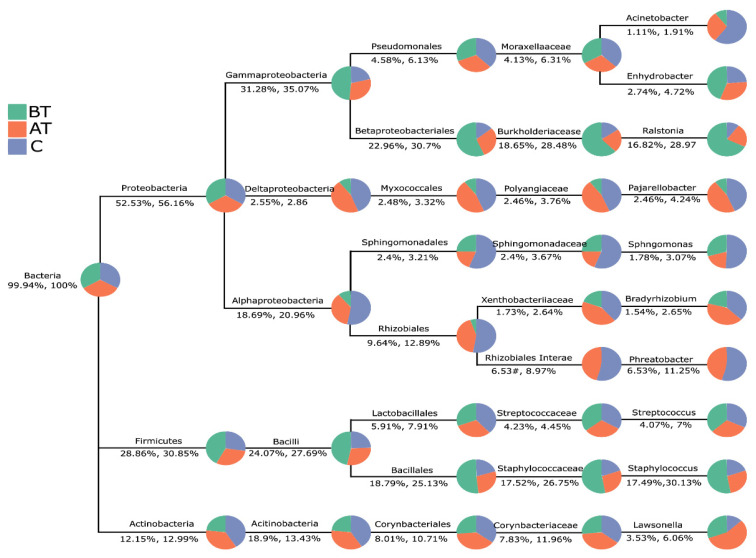
Specific taxonomic tree of facial microbiota (from phylum to genus) of the three experimental groups. Sectors with different colors in the pie chart represent different samples, and the size of the sector represents the relative abundance of the sample. The number below the classification name on the left side of the pie chart represents the average relative abundance (%) of all samples in this classification; the above number represents the total abundance.

**Figure 6 pharmaceuticals-16-00087-f006:**
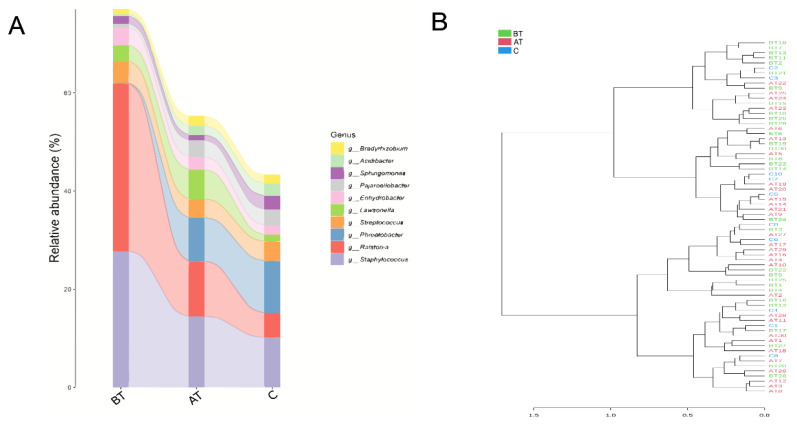
Difference between facial microbiota of the pre-treated (BT), treated (AT), and control (C) acne groups. (**A**): Sankey chart presenting the pattern of change in facial microbiota among the three experimental groups at the genus level. (**B**): Sample-wise similarity-based dendrogram of the three experimental groups.

**Figure 7 pharmaceuticals-16-00087-f007:**
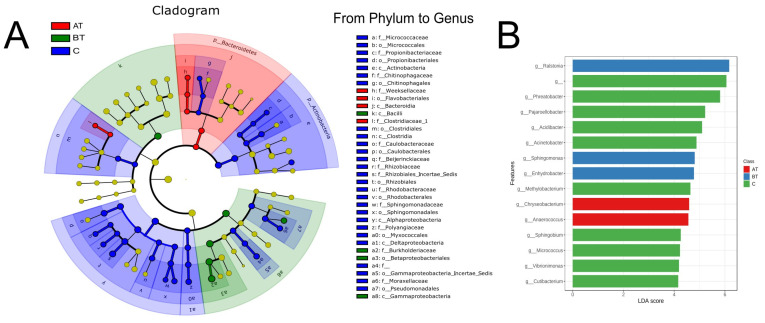
LEfSe analysis of the facial microbiota of pre-treated (BT), treated (AT), and control (C) acne groups. (**A)**: Cladogram representation of statistically consistent differences (Phylum—genus) in the three groups. Each circle’s diameter is in proportion to that taxon’s abundance. (**B**): Enriched genera (LDA > 3.5) in the three groups.

**Figure 8 pharmaceuticals-16-00087-f008:**
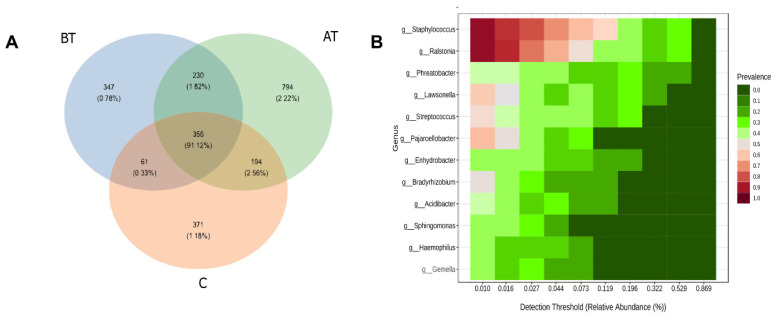
Core microbiota of the pre-treated (BT), treated (AT), and (C) acne groups. (**A**) presents the number and percent bacterial genera shared among the groups and unique to a certain group. (**B**) presents the percent of each genus in the core microbiota. The scale and color pattern are provided in the figure.

**Table 1 pharmaceuticals-16-00087-t001:** Total lesion count and GAGS score before and after treatment in the acne group.

Variable	BT	AT	*t*-Value	*p*-Value
Lesion count	39.67 ± 7.18	25.90 ± 8.40	8.06	<0.0001
GAGS scoring	14.20 ± 2.71	10.37 ± 3.12	7.95	<0.0001

BT: Before treatment; AT: After treatment.

**Table 2 pharmaceuticals-16-00087-t002:** Difference in microbial communities between the pre-treated (BT), treated (AT), and control (C) groups measured through Anosim and perMANOVA.

Constituencies	Index	*R*-Value	*p*-Value
Anosim analysis			
BT versus AT	Bray–Curtis	0.289	<0.001
BT versus C	Bray–Curtis	0.577	<0.001
AT versus C	Bray–Curtis	−0.028	0.6
BT versus AT versus C	Bray–Curtis	0.293	<0.001
perMANOVA analysis			
BT versus AT	Bray–Curtis	0.125	<0.001
BT versus C	Bray–Curtis	0.173	<0.001
AT versus C	Bray–Curtis	0.031	0.257
BT versus AT versus C	Bray–Curtis	0.150	<0.001

BT: Before treatment; AT: After treatment; C group: Control (Healthy).

## Data Availability

Sequence data are deposited to the NCBI SRA database under accession number PRJNA915114.

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
