# Peer review of "Stabilization of Acne Vulgaris-Associated Microbial Dysbiosis with 2% Supramolecular Salicylic Acid"

_pharmaceuticals, 2023, doi:10.3390/ph16010087_

Round 1

Reviewer 1 Report

This is very interesting article, thank You very much. I am waiting to get this product!

Add in the introduction :

https://www.ncbi.nlm.nih.gov/pmc/articles/PMC9653569/

https://www.ncbi.nlm.nih.gov/pmc/articles/PMC9318165/

https://www.ncbi.nlm.nih.gov/pmc/articles/PMC9267691/

1. Please write in 2.2. if the patients had any cosmetological treatments? What did they apply only this 2% SSA, what they used to clean the face? What about the make-up? Did they use any probiotics, yeast tablets, or sulfur tablets?

2. What were the exclusion criterias?

3. Next time I think the control group should be also like the clinical one (n=30)

Author Response

This is very interesting article, thank You very much. I am waiting to get this product!

Comment 1: Add in the introduction :
https://www.ncbi.nlm.nih.gov/pmc/articles/PMC9653569/
https://www.ncbi.nlm.nih.gov/pmc/articles/PMC9318165/
https://www.ncbi.nlm.nih.gov/pmc/articles/PMC9267691/
Response: Thank you so much for reviewing our article. The suggested changes are made to the revised version, and the given literature is added to the introduction section, reference numbers 10, 11, and 12.

Comment 2: Please write in 2.2. if the patients had any cosmetologically treatments? What did they apply only this 2% SSA, what they used to clean the face? What about the make-up? Did they use any probiotics, yeast tablets, or sulfur tablets?
Response: Thank you for mentioning these important points. Responses to all queries are added to the revised manuscript (section 4.1).
“Two months before sampling, the subjects and control had not taken any oral or topical antibiotics, glucocorticoids, probiotics, yeast or sulfur tablets, or any other cosmetologically treatment. All subjects were advised not to wear any make-up during treatment. The patients were treated with 2% SSA gel (twice a day, morning and evening) and a nourishing conditioning mask (once at night for 15 min). All subjects were advised to wash their faces 30 minutes before applying 2% SSA with water. None of the acne or control subjects were allowed to clean their face 24 h before sampling.”

Comment 3: What were the exclusion criteria?
Response: The exclusion criteria are added to the method section of the revised manuscript. The exclusion criteria included that; none of the subjects and control had a pregnancy, planning to have children, lactation, cosmetic or drug-induced acne, pro-inflammatory pigmentation, and scaring.

Comment 4: Next time I think the control group should be also like the clinical one (n=30).
Response: Thank you for your recommendation. Next time, the control group will also be like the clinical group. The current study selected 10 individuals with no facial acne as a control group. There was no statistical difference on the
bases of age and sex between the subjects and the control group (P > 0.05). The purpose of selecting 10 healthy individuals was to match their facial microbiota with acne patients before and after treatment.

Reviewer 2 Report

The manuscript by Hazrat Bilal et al. describes the impact on facial bacterioma of 2% SSA used in the treatment of acne.

This manuscript has many points that need to be revised if the article is to be publishable.

Global : 

Italicize "i.e.," "versus."

Passive tenses are preferred.

Methods: 

Paragraph 2.1 is redundant with part 2.4. Perhaps it is better to provide information on the therapeutic protocol used.

The criteria for selecting controls (healthy or simply non-acneic skin?) and patients (age of the disease) should be specified as well as the number of subjects required. Moreover, no statistics are necessary for the baseline data.

Give details on the extraction method. Is there any delay in the amount of extracted DNA introduced in the library preparation protocol? 

Were there negative and positive controls for microbiome analysis? If yes, please describe/ If no, please explain.

Which sequencing platform was used?

Give details of the versions used for the database and software. Also, the names of the bacteria must be updated.

Sequence data must be deposited on a public database.

Results : 

I recommend a re-numbering of the paragh (3.5 as 3.4.1 and 3.6 as 3.4.2 for example.

Table 2 and Figure 1 are redundant.

All explanations on the statistical methods are to be put in the corresponding methods sections.

Author Response

Comment 1: The manuscript by Hazrat Bilal et al. describes the impact on facial bacterioma of 2% SSA used in the treatment of acne.
This manuscript has many points that need to be revised if the article is to be publishable. Global : , Italicize "i.e.," "versus.", Passive tenses are preferred.
Response: Thank you for reviewing our manuscript. The manuscript is revised according to your given suggestions and recommendations.
Methods:

Comment 2: Paragraph 2.1 is redundant with part 2.4. Perhaps it is better to provide information on the therapeutic protocol used.
Response: Paragraph 2.1 (4.1 in the revised manuscript) is omitted. The information on the therapeutic protocol is provided in section 4.1 of the revised manuscript.

Comment 3: The criteria for selecting controls (healthy or simply non-acneic skin?) and patients (age of the disease) should be specified as well as the number of subjects required. Moreover, no statistics are necessary for the baseline data.
Response: The control group was of ten volunteers with no facial acne. The word “healthy” is omitted in the revised manuscript. The patient group comprised 30 individuals; all had moderate acne according to the Pillsbury grading system (grades II and III). The mean age of the patients was 22 years (range; 18-30), including 12 male and 18 female cases. The duration of acne was not noted because we selected the patients based on the GAGS score and lesion count, which is a globally accepted system [1]. Moreover, the statistic test for the baseline data was used because our number of acne patients and the control group differed. However, to show that based on their age and sex, there is no difference, we calculated the P-value, which was> 0.05. Hence the difference in numbers in the two groups did not affect our results.

Comment 4: Give details on the extraction method. Is there any delay in the amount of extracted DNA introduced in the library preparation protocol?
Response: The samples for molecular experiments were collected through swabs using the wetting solution (0.1% Tween-20 and 0.9% NaCl). The swabs were wiped up and down twenty times in the position of the face (6 × 6 cm2) that had multiple numbers of acne. The area from which the initial sampling
was taken was noted, and sampling was performed from the same position after treatment. For the control group, the cheeks region was selected for sample collection. All the procedures were performed in sterile conditions, and samples were immediately stored at -80 ºC until further processing.
Genomic DNA was extracted and purified from all the samples according to the kit manufacturer's protocol (Zymo Research Co., Ltd), following a preliminary step of bead-beating for 2 min. The integrity and concentration of DNA were detected following the Picogreen assay protocol [2]. The V3-V4 region of 16S rDNA was amplified using the primers pair; 515F:5´-GTGYCAGCMGCCGCGGTAA-3' and 806R:5´-GGACTACHVGGGTWTCTAAT-3´ and reaction conditions described earlier [3]. The amplified product was recovered using Zymoclean™ Gel DNA Recovery Kit; quantified by Qubit 2.0 Fluorometer (Thermo Scientific), and pooled in equimolar quantities. Subsequently, the DNA library was constructed using the NEBNext® Ultra™ II DNA Library Prep Kit for Illumina® (New England Biolabs (UK) Co., Ltd). High-throughput sequencing was performed using the illumina's HiSeq PE250 sequencing platform. The kit used for sequencing was HiSeq Rapid SBS Kit v2 (500 cycles).

Comment 5: Were there negative and positive controls for microbiome analysis? If yes, please describe/ If no, please explain.
Response: Current study aims to analyze the efficacy of 2% SSA on facial bacteria associated with acne vulgaris. For this purpose, 30 moderate acne vulgaris patients were selected as the experimental group. Also, 10 volunteers with no facial acne were selected as the control group. The microbiome of volunteers with no facial acne was taken as the positive control, and the microbe of acne patients before and after treatment was compared to them. In order to find out which specific microbial diversity was present in acne vulgaris samples and how they modulated after the treatment. Our results concluded that the acne vulgaris microbial diversity differed from the control group; however, it tends to be like the control group microbiota after treatment.

Comment 6: Which sequencing platform was used?
Response: High-throughput sequencing was performed using Illumina's HiSeq PE250 sequencing platform.

Comment 7: Give details of the versions used for the database and software. Also, the names of the bacteria must be updated.
Response: The software and tools used in the current study were SPSS (v.25), QIIME2 (2020.2), FLASH (1.2.11), R language (4.0.5), Python (3.7.4), and SILVA database (138). The updated names of bacteria are used in the current study.
Comment 8: Sequence data must be deposited on a public database.
Response: Sequence data are deposited to the NCBI SRA database under accession number PRJNA915114
Results :

Comment 8: I recommend a re-numbering of the paragh (3.5 as 3.4.1 and 3.6 as 3.4.2 for example.
Response: done accordingly

Comment 9: Table 2 and Figure 1 are redundant.
Response: Table 2 is omitted from the main text and added in the supplementary file (Table S1)

Comment 10: All explanations on the statistical methods are to be put in the corresponding methods sections.
Response: All the explanation on the statistical methods is in the corresponding method section.

Thank you..
References:
1. Doshi, A.; Zaheer, A.; Stiller, M.J. A comparison of current acne grading systems and proposal of a novel system. International journal of dermatology 1997, 36, 416-418, doi:10.1046/j.1365-4362.1997.00099.x.
2. Anantanawat, K.; Pitsch, N.; Fromont, C.; Janitz, C. High-throughput Quant-iT PicoGreen assay using an automated liquid handling system. BioTechniques 2019, 66, 290-294, doi:10.2144/btn-2018-0172.
3. Katz, R.; Ahmed, M.A.; Safadi, A.; Abu Nasra, W.; Visoki, A.; Huckim, M.; Elias, I.; Nuriel-Ohayon, M.; Neuman, H. Characterization of fecal microbiome in biopsy positive prostate cancer patients. BJUI compass 2022, 3, 55-61, doi:10.1002/bco2.104.

Reviewer 3 Report

The paper explores microbiota in acne patients before and after treatment with 2%SSA. Facial microbiota in healthy (I suppose acne-free) controls is measured for comparison.

The concept is interesting and the study is well performed. The results are promising.
I have some major comments regarding the way facts are presented or interpreted. The methods should be better described, recruiting and selection process. Actual limitations are not discussed and some comments are misleading. A true Discussion is lacking. Please see detailed comments and examples:

1. Methods: please define healthy individuals/volunteers. Healthy volunteers would mean no disease at all (not any medical or psychiatric condition). The patients and the controls are poorly described. How was 'healthy' defined? How were patients selected (i.e. any form of acne, only mild acne, who performed selection, etc..?) How were controls defined as healthy? If controls were individuals with no acne on the face, call them healthy individuals with no symptoms of facial acne.

2. Many of the statements in the introduction are non-nuanced, claiming full superiority of one treatment and complete uselessness of others. Claiming that 2%SSA is the 'advanced aproach for curing acne' is incorrect. Do you mean that a severe acne or acne fulminans should be treated with this advanced method and it is expected to be a cure? Your study only shows that SSA modifies the skin microbiome to become more diverse and more similar in non-acne skin. This is not enough to claim a cure.
Examples: You claim that 'other treatments are not practical due to limitations'. Any treatment for any disease would have some side effects or not be practical. it may be too strong to claim that conventional acne treatments cannot be used because of limitations. SSA can also cause irritation of the skin. Continuous use of emollients to prevent atopic eczema from flaring may be impractical, but no one would suggest that they not be used because of the limitations/impracticality of applying.
The way the text is formulated it may seem that salicylic acid is the only useful treatment for acne, only limited by its solubility.

3. Discussion: Please discuss bias when using a self-assessment grading. The no. of participants was small and some bias could be expected, the conclusions may therefore seem too strong. A true discussion is actually missing. The first part of the discussion belongs in the introduction. The second part is results. Later, again, text that belongs in the introduction is given with repetition of the results.
You can not conclude that the 2%SSA treatment significantly cured the acne, because the study does not allow for such a conclusion.

Minor:
- Abstract: First sentence: I suggest a more clear wording: The way the sentence is structured would mean that you measure the effect of 2%SSA also on healthy individuals. Or is it the effect on facial bacteria 2% SSA as compared to bacteria in normal skin?
 - Abstract: GAGS score is only given as an abbreviation. Any abbreviations should be written in full first time they are mentioned.
- In a scientific paper, it would be more correct to describe lesions as closed and open comedones instead of blackheads and whiteheads, papules instead of bumps, etc.
- In 2.1 Materials and instruments you write silicic acid (not salicylic): Do you mean silicic acid or is this a typo?
 - You write: 'disfigurement of patients' faces'. This may be more correctly changed to 'scars can change the skin's appearance, negatively, especially on visible areas, such as the face'. After all, scars on shoulders, back, or decolte will also be visible. It is also too strong to suggest that all individuals with some sequelae after acne are 'disfigured'.

Author Response

Reviewer 2

Comment 1: The paper explores microbiota in acne patients before and after treatment with 2%SSA. Facial microbiota in healthy (I suppose acne-free) controls is measured for comparison. The concept is interesting and the study is well performed. The results are promising. I have some major comments regarding the way facts are presented or interpreted. The methods should be better described, recruiting and selection process. Actual limitations are not discussed, and some comments are misleading. A true Discussion is lacking. Please see detailed comments and examples:

Response: Thank you for reviewing and appreciating our manuscript. All the suggestions and recommendations are made to the revised manuscript.

Comment 2: Methods: please define healthy individuals/volunteers. Healthy volunteers would mean no disease at all (not any medical or psychiatric condition). The patients and the controls are poorly described. How was 'healthy' defined? How were patients selected (i.e. any form of acne, only mild acne, who performed selection, etc..?) How were controls defined as healthy? If controls were individuals with no acne on the face, call them healthy individuals with no symptoms of facial acne.

Response: Thank you for mentioning this. The control group was comprised of ten volunteers with no facial acne. The patient group was selected according to the Pillsbury grading system; 30 patients having moderate acne (grades II and III) were selected from the department of dermatology at Sichuan Provincial People's Hospital, China [1]. The chief dermatologist Prof. Xinyu Lin (X.L) performed the patients' diagnosis and grading.

The changes are added to the revised manuscript (method section; 3.2)

Comment 3: Many of the statements in the introduction are non-nuanced, claiming full superiority of one treatment and complete uselessness of others. Claiming that 2%SSA is the 'advanced aproach for curing acne' is incorrect. Do you mean that a severe acne or acne fulminans should be treated with this advanced method and it is expected to be a cure? Your study only shows that SSA modifies the skin microbiome to become more diverse and more similar in non-acne skin. This is not enough to claim a cure.

Examples: You claim that 'other treatments are not practical due to limitations'. Any treatment for any disease would have some side effects or not be practical. it may be too strong to claim that conventional acne treatments cannot be used because of limitations. SSA can also cause irritation of the skin. Continuous use of emollients to prevent atopic eczema from flaring may be impractical, but no one would suggest that they not be used because of the limitations/impracticality of applying.

The way the text is formulated it may seem that salicylic acid is the only useful treatment for acne, only limited by its solubility.

Response: Thank you for pointing out our mistakes and detailed explanation of the therapeutic approaches to acne vulgaris. The introduction section is revised accordingly. Here I am mentioning the revised section;

“Some recent studies stated the promising anti-acne therapeutic efficacy of oral probiotics, oxybrasion, and synergistic treatment protocol of hydrogen purification and cosmetic acids [2-4]. However, these studies are preliminary, and further validations are required to prove their safety and efficacy. Currently, dermatologists use various treatment approaches, i.e., retinoids, antibiotics, anti-androgen, and salicylic acid, to cure acne. [5]. Salicylic acid is one of the peeling agents and is effective in treating acne due to its keratinolytic, anti-inflammatory, and bactericidal effects [6]. However, its application is limited due to its poor solubility in water and form crystals in low-pH alcoholic solutions, which cause skin irritation [7]. To overcome this limitation, a supramolecular technique based on a reversible and non-covalent bonding approach was applied to develop an intramolecular complex. This results in water-soluble supramolecular salicylic acid (SSA) which has the properties of slow releasing upon application, efficient in low-pH, and low skin irritation [8,9].

As mentioned earlier, microbial imbalance on the skin surface leads to acne production, and SSA is currently used for their treatment; however, their effect on acne-associated facial microbiota is not fully explored [8,10]. Therefore, in the current study, we aimed to evaluate the impact of 2% SSA on the composition and diversity of facial skin microbiota of acne patients using the high-throughput sequencing approach. The findings will provide information about the effect of 2% SSA on the facial-bacterial communities and new ideas for acne vulgaris treatment.”

Comment 4: Discussion: Please discuss bias when using a self-assessment grading. The no. of participants was small and some bias could be expected, the conclusions may therefore seem too strong. A true discussion is actually missing. The first part of the discussion belongs in the introduction. The second part is results. Later, again, text that belongs in the introduction is given with repetition of the results. You can not conclude that the 2%SSA treatment significantly cured the acne, because the study does not allow for such a conclusion.

Response: The discussion section is revised. Bias is discussed in the revised version, and study limitation is added at the end of the discussion section.

“The present study is the first of its kind that examines the effect 2% SSA on acne-associated facial microbiota. However, some limitations in the present study might be responsible for biased outcomes. Such as, only grade II and III, indicating moderate acne, were selected for evaluation of the 2% SSA effect, while grades I and IV indicating mild and severe acne, respectively, were not selected. Moderate acne was selected because it is more consistent and usually present in some form or another [11]. Similarly, only 30 acne patients were selected from only one hospital. Therefore, further studies are required to validate the effect of 2% SSA on facial microbiota in different acne types and geographical locations. “

Moreover, the repeted of results and extra text is omitted from the revised version, and more discussion is added (mentioned by track changes in the revised version). Similarly, the conclusion section is revised according to the given recommendation.

“Conclusion: The current study analyzed the facial bacteria of 30 moderate acne patients and their response after the treatment with 2% SSA. It is found that dysbiosis in facial microbiota can lead to acne development. The 2% SSA treatment significantly improved the GAGS score of moderate acne patients and very few adverse effects were noted in only two patients, which were recovered by themselves. The metagenomic analysis of the pre-treated and treated group concluded that the 2% SSA modulates the microbial diversity of patients toward like individuals with no facial acne. However, due to the relatively small sample size and selection of only moderate acne patients, this study suggests that more studies are necessary to draw a definitive conclusion on this topic.”

Minor comments

1- Abstract: First sentence: I suggest a more clear wording: The way the sentence is structured would mean that you measure the effect of 2%SSA also on healthy individuals. Or is it the effect on facial bacteria 2% SSA as compared to bacteria in normal skin?

Response: The first sentence of abstract is revised as: “Facial microbiota dysbiosis is an important factor in causing acne vulgaris. The present study aimed to analyze the effect of 2% Supramolecular Salicylic Acid (SSA) on acne-associated facial bacteria”.

 2- Abstract: GAGS score is only given as an abbreviation. Any abbreviations should be written in full first time they are mentioned.

Response: Abbreviation of GAGS score (global acne grading system) is added to the revised manuscript.

3- In a scientific paper, it would be more correct to describe lesions as closed and open comedones instead of blackheads and whiteheads, papules instead of bumps, etc.

Response: Thank you for your valuable suggestions, the signs of acne in are replaced in the revised manuscript by scientific terms.

4- In 2.1 Materials and instruments you write silicic acid (not salicylic): Do you mean silicic acid or is this a typo?

Response: Thank you for pointing out this; this was a typing error; the word is replaced by salicylic acid in the revised version.

5- You write: 'disfigurement of patients' faces'. This may be more correctly changed to 'scars can change the skin's appearance, negatively, especially on visible areas, such as the face'. After all, scars on shoulders, back, or decolte will also be visible. It is also too strong to suggest that all individuals with some sequelae after acne are 'disfigured'.

Response: The sentence is revised asSevere acne form scars, which can change the skin's appearance negatively, especially in visible areas, such as the face.”

Thank you..

References;

  1. Ramam, M. Skin: Clinical Dermatology. Indian journal of dermatology, venereology and leprology 2020, 86, 468, doi:10.4103/ijdvl.IJDVL_674_20.
  2. Sánchez-Pellicer, P.; Navarro-Moratalla, L.; Núñez-Delegido, E.; Ruzafa-Costas, B.; Agüera-Santos, J.; Navarro-López, V. Acne, Microbiome, and Probiotics: The Gut-Skin Axis. Microorganisms 2022, 10, doi:10.3390/microorganisms10071303.
  3. Chilicka, K.; Rogowska, A.M.; Szyguła, R.; Rusztowicz, M.; Nowicka, D. Efficacy of Oxybrasion in the Treatment of Acne Vulgaris: A Preliminary Report. Journal of clinical medicine 2022, 11, doi:10.3390/jcm11133824.
  4. Chilicka, K.; Rusztowicz, M.; Rogowska, A.M.; Szyguła, R.; Asanova, B.; Nowicka, D. Efficacy of Hydrogen Purification and Cosmetic Acids in the Treatment of Acne Vulgaris: A Preliminary Report. Journal of clinical medicine 2022, 11, doi:10.3390/jcm11216269.
  5. Ye, D.; Xue, H.; Huang, S.; He, S.; Li, Y.; Liu, J.; Wang, Z.; Zeng, W. A prospective, randomized, split-face study of concomitant administration of low-dose oral isotretinoin with 30% salicylic acid chemical peeling for the treatment of acne vulgaris in Asian population. International journal of dermatology 2022, 61, 698-706, doi:10.1111/ijd.16127.
  6. Al-Talib, H.; Al-Khateeb, A.; Hameed, A.; Murugaiah, C. Efficacy and safety of superficial chemical peeling in treatment of active acne vulgaris. Anais brasileiros de dermatologia 2017, 92, 212-216, doi:10.1590/abd1806-4841.20175273.
  7. Kligman, D.E.; Draelos, Z.D. Combination Superficial Peels With Salicylic Acid and Post-Peel Retinoids. Journal of drugs in dermatology : JDD 2016, 15, 442-450.
  8. Zheng, Y.; Yin, S.; Xia, Y.; Chen, J.; Ye, C.; Zeng, Q.; Lai, W. Efficacy and safety of 2% supramolecular salicylic acid compared with 5% benzoyl peroxide/0.1% adapalene in the acne treatment: a randomized, split-face, open-label, single-center study. Cutaneous and ocular toxicology 2019, 38, 48-54, doi:10.1080/15569527.2018.1518329.
  9. Berry, K.; Lim, J.; Zaenglein, A.L. Acne Vulgaris: Treatment Made Easy for the Primary Care Physician. Pediatric annals 2020, 49, e109-e115, doi:10.3928/19382359-20200211-01.
  10. Ramasamy, S.; Barnard, E.; Dawson, T.L., Jr.; Li, H. The role of the skin microbiota in acne pathophysiology. The British journal of dermatology 2019, 181, 691-699, doi:10.1111/bjd.18230.
  11. Eichenfield, D.Z.; Sprague, J.; Eichenfield, L.F. Management of Acne Vulgaris: A Review. Jama 2021, 326, 2055-2067, doi:10.1001/jama.2021.17633.

Round 2

Reviewer 2 Report

The manuscript has been rewritten according to most of my previous comments but some remain.

Comment 3: The criteria for selecting controls (healthy skin or just non-acne?) and patients (age of disease) should be specified as well as the number of subjects required. Also, no statistics are required for baseline data.

Response: The control group consisted of 10 volunteers without facial acne. The word "healthy" is omitted in the revised manuscript. The patient group included 30 individuals; all had moderate acne according to the Pillsbury classification system (grades II and III). The mean age of the patients was 22 years (range, 18-30), including 12 men and 18 women. The duration of acne was not noted because we selected patients based on the GAGS score and the number of lesions, which is a globally accepted system [1]. In addition, the statistical test for baseline data was used because our number of acne patients and the control group differed. However, to show that there was no difference according to their age and gender, we calculated the P value, which was> 0.05. Therefore, the difference in numbers in the two groups did not affect our results.

--> The numerical difference does not warrant statistical comparison. To be deleted.

Comment 5: Were there negative and positive controls for microbiome analysis? If yes, please describe/ If no, please explain.

Response: The current study is designed to analyze the efficacy of 2% SSA on facial bacteria associated with acne vulgaris. For this purpose, 30 patients with moderate acne vulgaris were selected as the experimental group. In addition, 10 volunteers without facial acne were selected as a control group. The microbiome of the volunteers without facial acne was taken as a positive control, and the microbiome of the acne patients before and after treatment was compared to it. In order to find out what specific microbial diversity was present in the acne vulgaris samples and how it modulated after treatment. Our results concluded that the microbial diversity of the acne vulgaris differed from that of the control group; however, it tended to resemble the microbiota of the control group after treatment.

--> it seems my previous comment was not understood correctly. I was talking about ANALYTICAL controls (sham community/blank sample). Were they considered for the microbiome analysis? If yes, please describe/ If no, please explain.

Author Response

Reviewer 2

Comment 1: The numerical difference does not warrant statistical comparison. To be deleted.

Response: Thank you for reviewing our manuscript. The numerical difference in the method section of the revised manuscript is omitted.

Comment 2: it seems my previous comment was not understood correctly. I was talking about ANALYTICAL controls (sham community/blank sample). Were they considered for the microbiome analysis? If yes, please describe/ If no, please explain.

Response: Thank you for explaining the comment. The reference standard is added to the method section of the revised manuscript.

In addition to the samples ZymoBIOMICSTM Microbial Community DNA Standard (D6305)https://www.zymoresearch.com/collections/zymobiomics-microbial-community-standards) was added as an internal reference. This mock community comprises the genomic DNA of eight bacterial species. For the experimental and analytical procedures, the same methods were used as for the samples; the bacterial compositions are shown in table S2 (Supplementary file).

Table S2: Bacterial composition of the ZymoBIOMICSTM Microbial Community DNA Standard expressed as 16S rRNA gene percentages.

Bacterial composition

Standard*(%)

Analysis**(%)

Bacillus subtilis

17.4

19.75

Staphylococcus aureus

15.5

14.95

Lactobacillus fermentum

18.4

16.65

Listeria monocytogenes

14.1

13.35

Escherichia coli

10.1

9.2

Salmonella enterica

10.4

9.55

Enterococcus faecalis

9.9

8.95

Pseudomonas aeruginosa

4.2

4.55

Footnote

*Community composition of the standard

**community composition after the actual analysis and testing of the standard.

Round 3

Reviewer 2 Report

The manuscript is nearly corrected.

Data are lacking about the negative analytical control.  Could the authors give more details ?

Author Response

Comment 1: Data are lacking about the negative analytical control.  Could the authors give more details?

Response: The reaction mixtures having ddH2O instead of samples were used as a negative control. No DNA bands were amplified for the negative control.